# Localization of *Frog Virus 3* Conserved Viral Proteins 88R, 91R, and 94L

**DOI:** 10.3390/v11030276

**Published:** 2019-03-19

**Authors:** Emily Penny, Craig R. Brunetti

**Affiliations:** Biology Department, Trent University, 1600 West Bank Dr, Peterborough, ON K9J 7B8, Canada; emily_penny@hotmail.com

**Keywords:** *Iridoviridae*, frog virus 3, FV3, ranavirus, immunofluorescence, intracellular localization

## Abstract

The characterization of the function of conserved viral genes is central to developing a greater understanding of important aspects of viral replication or pathogenesis. A comparative genomic analysis of the iridoviral genomes identified 26 core genes conserved across the family *Iridoviridae*. Three of those conserved genes have no defined function; these include the homologs of *frog virus 3* (FV3) open reading frames (ORFs) 88R, 91R, and 94L. Conserved viral genes that have been previously identified are known to participate in a number of viral activities including: transcriptional regulation, DNA replication/repair/modification/processing, protein modification, and viral structural proteins. To begin to characterize the conserved FV3 ORFs 88R, 91R, and 94L, we cloned the genes and determined their intracellular localization. We demonstrated that 88R localizes to the cytoplasm of the cell while 91R localizes to the nucleus and 94L localizes to the endoplasmic reticulum (ER).

## 1. Introduction

The viral family *Iridoviridae* are large (~120–200 nm in diameter) icosahedral dsDNA viruses that replicate in the cytoplasm and nucleus of infected cells [1]. Iridoviral genomes are terminally redundant and circularly permuted [2,3,4]. Iridoviruses are known to infect invertebrates as well as poikilothermic vertebrates such as: fish, amphibians, and reptiles [5]. Members of the *Iridoviridae*, some of which exhibit very high pathogenicity, have been implicated in significant infection events in wild and captive populations, leading to considerable impacts on wildlife conservation, fish farming, and aquaculture [6]. The family *Iridoviridae* is currently divided into two subfamilies the *Alphairidovirinae* and *Betairidovirinae*. The *Alphairidovirinae* is composed of three genera *Lymphocystivirus, Megaloctyivirus*, and *Ranavirus* while the *Betairidovirinae* is composed of the *Chloriridovirus* and *Iridovirus* genera [7,8].

A comparative genomic analysis of members of the family *Iridoviridae* demonstrated that there are 26 open reading frames (ORFs) that are conserved across all members of the family [9]. While novel genes generally define the unique aspects of the life cycle of a particular virus species, conserved genes are usually related to or play a role in essential mechanisms of the viral life cycle [9,10,11,12,13,14,15,16]. There are a number of functions that conserved genes are commonly involved in including: RNA and DNA synthesis and modification, viral protein processing, and viral structure and assembly [9,10,11,12,13,14,15,16]. The characterization and further understanding of the function of conserved viral genes is central to developing a greater understanding of important aspects of viral replication or pathogenesis.

*Frog virus 3* (FV3) is the type species of the genus *Ranavirus* and is the most extensively studied iridovirus at the molecular level [6]. The FV3 genome, excluding the terminal redundancy, is 105,903 base pairs long and encodes 98 non-overlapping predicted ORFs, which range in size from 50 to 1293 amino acids in length [15]. Although 26 genes are conserved across the family *Iridoviridae* [9], the products of three of the identified core genes do not appear to have a fully characterized function; these include FV3 ORFs 88R, 91R, and 94L [15]. Though the proteins encoded by these 3 ORFS have yet to be fully characterised within the family *Iridoviridae*, it is likely that their functions are essential based on their highly conserved nature among the iridoviruses.

Sequence homology provides some insight into potential roles for these genes. FV3 88R has similarity to ERV1/ALR (augmenter of liver regeneration) family of proteins [15]. In mammals, ALR is involved in the reduction of renal damage in liver transplant recipients [17]. In viruses, homologues of the ERV1/ALR family of proteins are highly conserved, though poorly characterized [18,19]. Viral ERV1/ALR homologues are thiol oxidoreductases; which are involved in the formation of disulfide bonds and are usually contained in the endoplasmic reticulum (ER) of normal eukaryotic cells [18,20].

In addition to 88R, basic information can be obtained from FV3 91R and 94L sequences. 91R has sequence similarity to immediate-early protein ICP-46 [15]. The FV3 gene 94L has a KKXX-like ER retrieval motif that was identified at the C-terminal of 94L (LRKV). A KKXX motif is responsible for recycling protein from the Golgi complex to the ER [21] suggesting that 94L might be localized to the secretory pathway.

In this paper, we propose to begin to characterize these genes by cloning the putative genes and determining the site of intracellular accumulation of the proteins by ectopic expression and microscopic analysis.

## 2. Materials and Methods

### 2.1. Isolation of FV3 DNA

Fathead minnow (FHM) (ATCC CCL-42) cells grown to approximately 80% confluence were infected with FV3 (American Type Culture Collection, Manassas, VA, USA) at a multiplicity of infection (MOI) of 0.1. When cytopathic effects (CPE) were observed, the cells were harvested and resuspended in 400 µL of water. The cells were freeze (−80 °C)-thawed 3 times. An equal volume of phenol/chloroform was added to the cells and after mixing by vortexing, the aqueous phase was transferred to a fresh tube. The phenol/chloroform extraction step was repeated twice more. Following transfer, 10% (*v*/*v*) 5 M sodium acetate and 200% (*v*/*v*) ethanol (100%) was added to the tube and the mixture was incubated on ice for 15 min. The viral DNA was then spun at 10,000× *g* for 10 min. The pellet of viral DNA was air dried and resuspended in 50 µL of water and stored at −20 °C.

### 2.2. Cloning FV3 88R, 91R, and 94L into pGEM-T Easy Vector

Gene specific oligonucleotide primers were designed for FV3 ORFs 88R, 91R, and 94L. The primers were designed to add a *Hind*III restriction enzyme site at the 5’ end of each gene and a *Xho*I restriction enzyme site at the 3’ end of each gene (Invitrogen, Burlington, Canada) (Table 1). PCR was then performed using these gene specific primers. In a PCR reaction tube, 1× PCR buffer (Invitrogen, Burlington, Canada), 3 mM MgCl_2_ (Invitrogen, Burlington, Canada), 2.5 units of TAQ DNA polymerase at 5 units/µL (Invitrogen, Burlington, Canada), 0.2 mM dNTPS, 0.1 mM gene specific forward and reverse primers (Table 1), 1 µg FV3 DNA and water to a final volume of 50 µL. The PCR reaction was then placed in a thermocycler for 30 cycles of 94 °C for 30 s, 52 °C for 30 s and 72 °C for 2 min. The PCR product, amplified DNA encoding each target sequence, was ligated into the pGEM-T Easy vector (Promega, Madison, WI, USA) as per the manufacturer’s protocol. The reaction was then stored at 4 °C overnight to ensure optimal ligation. Plasmid vectors were then transformed into DH5-α *E. coli* cells (Invitrogen, Burlington, Canada) following the manufacturer’s protocol and plated on a LB agar plates prepared with 10 mg/mL tryptone, 5 mg/mL yeast extract, 10 mg/mL NaCl, 1.5% agar and 0.1% ampicillin (50mg/mL). Colonies were isolated and screened for the insert.

### 2.3. Cloning of FV3 88R, 91R, and 94L into a Eukaryotic Expression Vector

Bacteria containing the pGEM T-easy 88R, 91R, or 94L was grown in LB media (10 mg/mL tryptone, 5 mg/mL yeast extract, 10 mg/mL NaCl) containing 100 µg/mL ampicillin. Plasmid DNA from the bacteria was extracted using a PureLink HiPure Plasmid DNA Purification Kit (Invitrogen, Burlington, Canada) as per the manufacturer’s protocol. A restriction enzyme digest was performed on the plasmid DNA to extract the DNA encoding each ORF of interest and the expression vector pcDNA3.1. Briefly, 20 µg of pGEM-T Easy plasmid DNA or pCDNA3.1 was digested with 25 units *Xho*I at 10 U/µL (Invitrogen, Burlington, Canada) and 25 units of *Hind*III at 10 U/µL (Invitrogen, Burlington, Canada), 1× React 2 reaction buffer, and water to a final volume of 70 µL. The reaction was incubated at 37 °C overnight. The product of the restriction enzyme digest had 14 µL of gel electrophoresis 20% loading dye added to the reaction tube and the entire reaction was run on a 1% agarose gel. The fragments of DNA corresponding with each ORF of interest and pcDNA3.1 were then excised from the gel and the DNA was extracted from the agarose gel using the QIAEX II gel extraction kit (Qiagen, Mississauga, Canada) as per the manufacturer’s protocol. The product of the gel extraction for each gene of interest was then ligated into the pcDNA3.1 expression vector using a high efficiency T4 DNA ligase (Invitrogen, Burlington, Canada) as per the manufacturer’s protocol. The ligation of each of the genes of interest into pcDNA3.1 created the expression constructs containing the gene of interest in frame with a myc tag and are called pcDNA3-88R-myc, pcDNA3-91R-myc, and pcDNA3-94L-myc.

### 2.4. Transfection of Eukaryotic Expression Vector into Eukaryotic Cells

Baby Green Monkey Kidney (BGMK) cells (BGMK-DAF P17, American Type Culture Collection, Manassas, VA, USA) were cultured at 37 °C in Dulbecco’s Modified Eagle’s Medium (DMEM; Gibco, Burlington, Canada), which is supplemented with 7% (*v*/*v*) heat-inactivated fetal bovine serum, 1% (*v*/*v*) l-glutamine and 1% (*v*/*v*) penicillin-streptomycin, on glass cover slips in a 6-well plate, to 90–95% confluence. Eukaryotic expression vectors, pcDNA3-88R-myc, pcDNA3-91R-myc, or pcDNA3-94L-myc, were transfected into cells using Lipofectamine 2000 (Invitrogen, Burlington, Canada). Specifically, 4 µg of DNA was diluted in 250 µL of serum-free DMEM (1% (*v*/*v*) l-glutamine, 1% (*v*/*v*) penicillin-streptomycin) and mixed gently. 10 µL of Lipofectamine 2000 was diluted in 250 µL of serum-free DMEM and incubated for 5 min. The diluted DNA and diluted Lipofectamine 2000 were combined, for a total volume of 500 µL, mixed gently and incubated at room temperature for 20 min. The 500 µL of complexes was then added to each well, containing cells and 2 mL supplemented DMEM growth medium and mixed by gently rocking the plate back and forth. The transfected cells were incubated at 37 °C for 48 h.

### 2.5. Indirect Immunofluorescence

The transfected cells were fixed and permeabilized 48 hours post-transfection. Specifically, the wells were each washed twice in 2 mL of phosphate buffered saline solution (Gibco, Burlington, Canada) with a pH of 7.4 for two minutes each wash. 1 mL of a 3.7% paraformaldehyde solution in PBS was applied to each well for 10 min to fix the cells. Upon removal of the paraformaldehyde solution, the wells were washed twice in 2 mL of PBS for 2 min each wash. The cells were then incubated with 1 mL of a 0.1% Triton X-100 in PBS for 4 min. The cells were then washed twice in 2 mL of PBS. The cells were then incubated overnight at 4 °C with 2 mL of block buffer (50 mM Tris HCl (pH 6.8), 150 mM NaCl, 0.5% (*v*/*v*) NP-40, 5.0 mg/mL BSA). The block buffer was removed and the wells were washed twice with 2 mL of Wash Buffer (50 mM Tris HCl (pH6.8), 150 mM NaCl, 0.5% (*v*/*v*) N-40, 1 mg/mL BSA). The cover slips were removed from the well and the primary antibodies were applied to the cells and incubated for 60 min at room temperature. The primary antibodies used were rabbit anti-myc (Sigma, Saint Louse, USA) diluted to 1:400 in wash buffer, mouse anti-c-myc (Roche, Mississauga, Canada) diluted to 1:400 in wash buffer, and rabbit anti-protein disulfide isomerase (PDI) (Sigma, Saint Louis, MO, USA) diluted to 1:500 in wash buffer. The cells were then washed 3 times with Wash Buffer. Secondary antibodies goat anti-mouse conjugated Cy3 (Jackson ImmunoResearch Laboratories, Inc., West Grove, PA, USA) diluted to 1:200 in Wash Buffer and goat anti-rabbit conjugated fluorescein (FITC) (Jackson ImmunoResearch Laboratories, Inc., West Grove, USA) diluted to 1:200 in Wash Buffer. The cells were incubated at room temperature for 60 min and were washed 3 times with Wash Buffer. The cells were then incubated with TO-PRO-3 at a dilution of 1:8000 in water (Molecular Probes, Eugene, OR, USA) for 7 min. The cells were washed 3 times in Wash Buffer and then mounted on slides using Vectashield (Vector Laboratories, Burlingame, CA, USA). The slides were then visualized using a Leica TCS SP2 SE confocal microscope. The localization of each gene was confirmed in at least 3 independent transfections. Multiple examples of each gene were visualized, and results were all consistent. The sample images shown are representative and consistent with the staining pattern observed for each gene.

## 3. Results

### 3.1. FV3 ORF 88R Localize to the Cytoplasm of Transfected Cells

To determine the site of 88R localization, BGMK cells were transfected with pcDNA3-88R-myc and the proteins were visualized by indirect immunofluorescence. The expression of 88R was found to be distributed throughout the cytoplasm (Figure 1). In particular, the staining of 88R was punctate, seeming to be evenly distributed throughout the cell (Figure 1). We examined many examples of 88R staining and they all shared similar cytoplasmic staining.

### 3.2. FV3 ORF 91R Localizes to the Nucleus of Transfected Cells

FV3 91R is another protein known to be conserved across the family *Iridoviridae*. BGMK cells were transfected with pcDNA3-91R-myc and proteins visualized by indirect immunofluorescence. Figure 2 demonstrates that 91R colocalizes with the TO-PRO-3 staining demonstrating that 91R localizes to the nucleus. Interestingly, there are pockets within the nucleus that lack both 91R and TO-PRO-3 expression (white arrows in Figure 2). Since TO-PRO-3 binds to DNA, it is interesting that 91R mimics the expression pattern suggesting that it localizes to the areas of the nucleus where the DNA resides.

### 3.3. FV3 ORF 94L Localizes to the ER in Transfected Cells

To determine the site of intracellular accumulation of 94L, BGMK cells were transfected with pcDNA3-94L-myc. Forty-eight hours post-transfection, the cells were fixed and indirect immunofluorescence was used to visualize 94L-myc protein expression. FV3 94L exhibited strong expression around the nucleus (Figure 3). This perinuclear localization correlated with PDI expression, a marker of the ER [22] (Figure 3). Although the correlation between PDI and 94L expression was not complete, the data does show overlap between the marker and 94L (Figure 3).

## 4. Discussion

The characterization of conserved viral proteins is necessary to gain further insight into the mechanisms of viral infection and replication. Conserved proteins are essential to the viral infection cycle, as their presence in the genomes of multiple members of the same viral family suggest. The purpose of this study was to begin the characterization of the protein products of three FV3 ORFs: 88R, 91R, and 94L.

Cytoplasmic localization was observed for FV3 88R. FV3 88R has similarity to ERV1/ALR family of proteins [15]. Interestingly, the family *Poxviridae* contains a cytoplasmic disulfide bond formation pathway that involves vaccinia virus E10, A2.5, and G4 [18,20]. The vaccinia virus E10 protein is an ERV1/ALR homologue that functions in the cytoplasm as part of the poxvirally encoded cytoplasmic pathway of disulfide bond formation [18,20,23,24]. The localization pattern exhibited by 88R is consistent with the cytoplasmic localization of viral ERV1/ALR homologues [18,20]. Also, it is known that ERV1/ALR homologues are highly conserved in both the *Poxviridae* and *Afarviridae* families, which are some of the most closely related viral families to the family *Iridoviridae* [20,25]. It is possible that FV3 88R performs a similar function to vaccinia virus E10 and there may also be the potential for the identification of a cytoplasmic pathway of disulfide bond formation in iridoviruses.

The second core gene examined, FV3 91R, localized to the nucleus of transfected cells. 91R has sequence similarity to immediate-early protein ICP-46 [15]. Interestingly, transcriptome analysis of FV3 suggested that 91R is an immediate-early gene [26]. FV3 is known to initiate genomic replication early in the viral replication cycle in the nucleus of infected cells [3]. There is a possibility that 91R is involved in an early event in FV3 replication which occurs in the nucleus of infected cells immediately following viral entry. Although the function of 91R may not be limited to being involved in the early events of viral replication. For example, the nuclear localization of 91R might also suggest that 91R regulates viral gene transcription. Therefore there are a number of areas that 91R could function in the nucleus and more research is required to understand the nuclear function of the gene.

The FV3 94L gene colocalized with the ER marker PDI. Many viruses use the ER and secretory pathway [27,28,29]. For example, both vaccinia virus and African swine fever virus (ASFV) encode viral proteins that are involved in the modulation of the ER and secretory pathway to accommodate viral replication [27,28,29]. As previously noted, a KKXX-like ER retrieval motif was identified at the C-terminal of 94L (LRKV). A KKXX motif is responsible for recycling protein from the Golgi complex to ER [21]. Retrieval motifs have been found to modulate secretory pathways and it has been experimentally determined that deletion mutant proteins lacking the double lysine motif are transported downstream to the vesicle [21]. The presence of a KKXX-like motif at the C-terminal of 94L further supports the role for 94L in the viral modulation of the secretory pathways of the host cell and adds further support to a role for 94L in the ER. Future research could delete the ER retrieval motif from 94L to see if the localization of the gene is altered.

One caveat of the experiments described is that the genes are ectopically expressed in non-host cells. We used the mammalian cell line, BGMK for a number of reasons. For localization studies, it is much easier to use mammalian cell lines, where a variety of intracellular organelle markers exist. In addition, we have used BGMK cells in the past to localize a variety of FV3 genes, and have found that the localization in BGMK cells mimics the localization in amphibian cell lines [30,31]. It is noteworthy that BGMK cells are infectible by FV3 when the cells are incubated at 30 °C [32]. Although this work was done in a mammalian cell line, it is always possible that the localization of the gene could be altered if the target gene interacts specifically with amphibian proteins. In addition, ectopic expression often results in over-expression of the target protein which may overwhelm intracellular targeting mechanisms resulting in aberrant localization. Finally, it is important to remember that these genes are being expressed outside of an FV3 infection. If 88R, 91R, or 94L interact with other viral proteins, this could also result in altered localization of the gene in and FV3-infected cell.

The results of cellular localization experiments have identified the subcellular expression patterns of the three highly conserved, uncharacterized FV3 proteins included in this study. Based on the results of the localization experiments, possible functions have been assigned to each protein, but these functions are speculative and extensive work is required to describe the actual functions of any of the proteins in an FV3 infection. The further characterization of FV3 88R, 91R, and 94L will provide insight into the cellular cycle of FV3 infection and maybe even certain cellular processes that are modified or commandeered by FV3 in order to successfully infect the host.

## Figures and Tables

**Figure 1 viruses-11-00276-f001:**
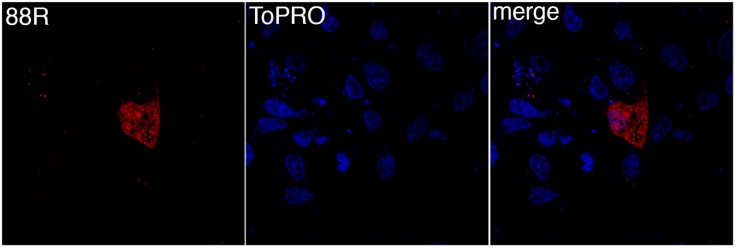
FV3 88R localizes to the cytoplasm. Baby Green Monkey Kidney (BGMK) cells were transfected with pcDNA3-88R. 48 hours post transfection, cells were fixed, and indirect immunofluorescence was performed using rabbit anti-myc antibodies (red) and TO-PRO-3 (blue). Images were captured at 100× magnification using a confocal microscope.

**Figure 2 viruses-11-00276-f002:**
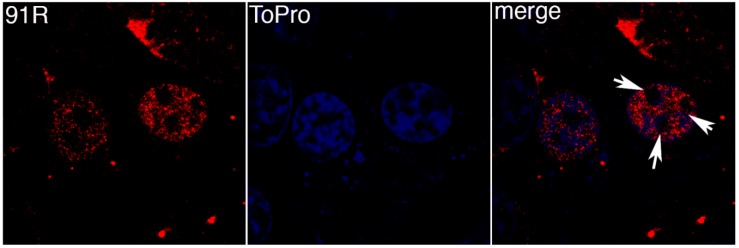
91R localizes to the nucleus. BGMK cells were transfected with pcDNA3-91R-myc. 48 hours post-transfection, the cells were fixed and indirect immunofluorescence was performed using rabbit anti-myc antibodies (red) and TO-PRO-3 (blue). Images were captured at 100× magnification using a confocal microscope. White arrows highlight nuclear areas that lack 91R and TO-PRO-3.

**Figure 3 viruses-11-00276-f003:**
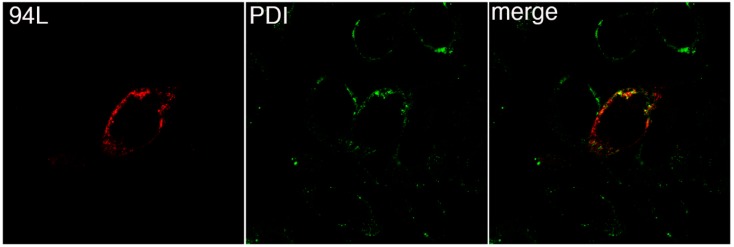
94L localizes to the Endoplasmic Reticulum. BGMK cells were transfected with pcDNA3-94L-myc. Forty-eight hours post-transfection, the cells were fixed and indirect immunofluorescence was performed using mouse anti-myc antibodies (red) and rabbit anti-PDI antibodies (green). Images were captured at 100× magnification using a confocal microscope.

**Table 1 viruses-11-00276-t001:** Oligonucleotide sequences designed to isolate the conserved genes from the *frog virus 3* (FV3) genome. Restriction sites were added 3 base pairs upstream of the start codon (*Hind*III—AAGCTT) and in the place of the stop codon (*Xho*I—CTCGAG) was added at the 3’ end of each gene.

Name	Sequence
88R-F	5′-AAGCTTAAAATGCACGGTTGCAATTG-3′
88R-R	5′-CTCGAGGTTAAAAGTGCTCGTATTTG-3′
91R-F	5′-AAGCTTAACATGGCAAACTTTGTGAC-3′
91R-R	5′-CTCGAGGGCTCTGACCACAAACAG-3′
94L-F	5′-AAGCTTGCAATGGATCCAGAAGGAATG-3′
94L-R	5′-CTCGAGCAGCACCTTTCTCAGGTAC-3′

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
