# Peer review of "Localization of Frog Virus 3 Conserved Viral Proteins 88R, 91R, and 94L"

_viruses, 2019, doi:10.3390/v11030276_

Round 1

Reviewer 1 Report

The manuscript, “Characterization of conserved viral proteins from the family Iridoviridae” by Penny and Brunetti describe experiments to determine the cellular localization of 3 ectopically expressed Frog virus 3 (FV3) proteins. The authors examined three highly conserved ranavirus genes in baby green monkey kidney (BGMK) mammalian cells. The cellular location of these ectopically expressed proteins was then used to begin to characterize the function of these conserved FV3 genes. The manuscript is well written and the data are presented in a logical manner. However, there are some issues that need to be addressed.

The title is misleading. The authors have only preformed an ectopic expression and cellular localization experiment at one time point in one cell line. This limited approach does not warrant gene “characterization”. Therefore, the title needs modifying to more accurately represent the study presented, unless further experimentation is performed to characterize the function of these FV3 genes.

The data presented are based on ectopic expression experiments in a mammalian cell line. Why were the BGMK cells chosen? Why were experiments not also performed in a cell line, and temperature, that supports replication of FV3 (e.g. FHM cells that were used to obtain FV3 DNA)? How does ectopic expression of FV3 genes at a temperature that is not permissive for FV3 replication influence the cellular localization data observed? Why was only a 48 hr time point used to assess cellular localization, especially as two immediate early FV3 genes (ORFs 91 and 94), were examined? The manuscript needs more of a discussion about ectopic expression, and the limitations of these experiments, especially in a cell line that is not permissive for FV3 growth and replication.

It is unclear how many cells were analyzed or how many times the experiment was repeated for each gene. This is critical information that is missing in the current version of the manuscript. That said, on lines 133-134 the authors state, “We examined many examples 88R staining….” but it is unclear quantitatively what this means for this gene and no other information about the other 2 genes is included in the manuscript. More information about the repeatability of the data presented is required.

The Discussion section is disappointing. The FV3 genes used in this study had identified signatures/domains prior to performing the localization experiments. This information should have been included in the Introduction section of the manuscript. As written, it appears that the 3 FV3 genes analyzed had no known signature/domain until after ectopic expression analysis. It makes more sense to present this information and make hypotheses about cellular location in the Introduction. The Discussion section can then expand on the data obtained and present thoughts for gene function.

Table 1 is missing from the manuscript.

Minor comments

Paragraph 1 of the Introduction: The appropriate way to reference a genus or family is to use the convention, genus Ranavirus and family Iridoviridae, not Ranavirus genus or Iridoviridae family (same for lines 40, 141, 190, 191). In addition, the family Iridoviridae has now been divided into two subfamilies. Please include the most up to date taxonomy information in the manuscript.

line 29: Suggest changing “were” to “are” (two times on this line).

line 36: Frog virus 3 should be in italics as it is a recognized species.

line 45: Suggest adding, “….by ectopic expression and microscopic analysis.”

line 49: Is there an ATCC number for the FHM cells?

line 58: Suggest ending the sentence after “….94L.” Then start the next sentence.

line 62: Change “unites” to “units”.

line 71-72: Please define or describe “positive colonies” that were isolated.

line 74: How is DNA “grown up”? Please clarify.

line 91: Is there an ATCC # for BGMK cells. Also, what temperature was used to grow and transfect these cells?

line 159: Suggest changing “…perinuclear expression…” to “…perinuclear localization….”

Author Response

The manuscript, “Characterization of conserved viral proteins from the family Iridoviridae” by Penny and Brunetti describe experiments to determine the cellular localization of 3 ectopically expressed Frog virus 3 (FV3) proteins. The authors examined three highly conserved ranavirus genes in baby green monkey kidney (BGMK) mammalian cells. The cellular location of these ectopically expressed proteins was then used to begin to characterize the function of these conserved FV3 genes. The manuscript is well written and the data are presented in a logical manner. However, there are some issues that need to be addressed.

The title is misleading. The authors have only preformed an ectopic expression and cellular localization experiment at one time point in one cell line. This limited approach does not warrant gene “characterization”. Therefore, the title needs modifying to more accurately represent the study presented, unless further experimentation is performed to characterize the function of these FV3 genes.

Response: We agree.  We have changed the title, changing the word “Characterization” to “Localization”.  We have also made some additional adjustments to the title to more accurately reflect the scope of the manuscript. 

The data presented are based on ectopic expression experiments in a mammalian cell line. Why were the BGMK cells chosen? Why were experiments not also performed in a cell line, and temperature, that supports replication of FV3 (e.g. FHM cells that were used to obtain FV3 DNA)? How does ectopic expression of FV3 genes at a temperature that is not permissive for FV3 replication influence the cellular localization data observed? Why was only a 48 hr time point used to assess cellular localization, especially as two immediate early FV3 genes (ORFs 91 and 94), were examined? The manuscript needs more of a discussion about ectopic expression, and the limitations of these experiments, especially in a cell line that is not permissive for FV3 growth and replication.

Response: We have added a paragraph in the Discussion section to address these issues (lines 295-309).  In the paragraph, we go beyond the reviewers request and also talk about the limits of localization studies in the absence of a viral infection. Specifically, we acknowledge the caveats of working in a non-host species but also describe some of the benefits of working in mammalian systems.  However, we have used BGMK cells in the past to do localization studies with FV3 genes (references included) as well as showing that BGMK cells are infectible by FV3 at 30°C.  We do not do localization studies in BGMK cells at 30°C as BGMK cell morphology is altered when they are grown at 30°C (described in Discussion). 

It is unclear how many cells were analyzed or how many times the experiment was repeated for each gene. This is critical information that is missing in the current version of the manuscript. That said, on lines 133-134 the authors state, “We examined many examples 88R staining….” but it is unclear quantitatively what this means for this gene and no other information about the other 2 genes is included in the manuscript. More information about the repeatability of the data presented is required.

Response: Within the Materials and Methods, we added the statement (lines 195-197): “The localization of each gene was confirmed in at least 3 independent transfections.  Multiple examples of each gene were visualized, and results were all consistent. The sample images shown are representative and consistent with the staining pattern observed for each gene”. Each gene was actually tested with multiple intracellular organelle markers as part of the localization process. Much of that data isn’t shown as it is isn’t informative to show that a gene doesn’t co-localize with a particular marker.  However, multiple experiments were performed, and the results were consistent with each gene. The above statement hopefully makes clear that we haven’t “cherry-picked” best images but are showing data reflective of the samples examined.

The Discussion section is disappointing. The FV3 genes used in this study had identified signatures/domains prior to performing the localization experiments. This information should have been included in the Introduction section of the manuscript. As written, it appears that the 3 FV3 genes analyzed had no known signature/domain until after ectopic expression analysis. It makes more sense to present this information and make hypotheses about cellular location in the Introduction. The Discussion section can then expand on the data obtained and present thoughts for gene function.

Response: We agree.  We moved two paragraphs of material from the Discussion to the Introduction (lines 82-92).  We then focused our discussion to include how our data confirmed or expanded upon the inferences from the sequence data.  

Table 1 is missing from the manuscript.

Response: Table 1 was added to the Materials and Methods section.

Minor comments

Paragraph 1 of the Introduction: The appropriate way to reference a genus or family is to use the convention, genus Ranavirus and family Iridoviridae, not Ranavirus genus or Iridoviridae family (same for lines 40, 141, 190, 191). In addition, the family Iridoviridae has now been divided into two subfamilies. Please include the most up to date taxonomy information in the manuscript.

Response: changes made throughout the document

line 29: Suggest changing “were” to “are” (two times on this line).

Response: changes made.

line 36: Frog virus 3 should be in italics as it is a recognized species.

Response: change made.

line 45: Suggest adding, “….by ectopic expression and microscopic analysis.”

Response: change made.

line 49: Is there an ATCC number for the FHM cells?

Response: added to the Materials and Methods

line 58: Suggest ending the sentence after “….94L.” Then start the next sentence.

Response: change made.

line 62: Change “unites” to “units”.

Response: Fixed.

line 71-72: Please define or describe “positive colonies” that were isolated.

Response: Line has been adjusted to reflect colonies were isolated and screened for the insert.

line 74: How is DNA “grown up”? Please clarify.

Response: We have expanded the section to describe how we grow up bacteria in LB/ampicillin media and then how we extract the plasmid DNA.

line 91: Is there an ATCC # for BGMK cells. Also, what temperature was used to grow and transfect these cells?

Response: Changes made.

line 159: Suggest changing “…perinuclear expression…” to “…perinuclear localization….”

Response: Changes made.

Reviewer 2 Report

Characterization of Conserved Viral Proteins from the family Iridoviridae

By Penny and Brunetti

viruses-453738

This manuscript represents an interesting addition to the limited knowledge we have on genome evolution in Iridoviviridae, and in particular the function of conserved genes in this family. The manuscript reads well and while some parts are redundant and could benefit from a bit of trimming, it presents interesting avenues for future work and functional exploration of these genes. Additionally, even though most of the readers would have the basic cell biology background to understand the many acronyms used, I think some needs to be defined, especially when they represent important findings as opposed to commonly used chemicals or biological agents.

Here are some minor editorial suggestions:

-Page 1, line 12: define ORF [while on line 29, add ORF after open reading frames]

-Page 1, line 17: define ER

-Page 3, line 110: I suggest “The cells were then incubated overnight at 4C with 2ml of block buffer…”

-Page 3, line 122: remove “then the cells”

-Page 3, line 123: remove “and the cells were incubated”

-Page 3, line 129-130: this sentence should be included in the Methods unless the format of a Brief Report is more flexible

-Page 4, line 141-143: these sentences are not necessary here and at best would need to be in the Methods, unless the format of a Brief Report is more flexible

-Page 4, line 145: To-PRO-3 should be defined in the Methods

-Page 5, line 161: given the figure, I find “strong overlap” a bit exaggerated

-Page 5, line 178: replace the second “thiol oxidoreductases” by “which are”

-Page 5, line 177-187: this whole section could be reduced as it goes into much details that are possibly out of scope for this Brief Report

-Page 5, line 199: “may not be limited”

-Page 5, line 199-200: why is the basis for this particular function? Should be justified

-Page 6, line 201-204: these two sentences could be merged and shortened

-Page 6, line 217: I suggest “in order to a successful infection of the host”

Best regards,

David Lesbarrères

Author Response

This manuscript represents an interesting addition to the limited knowledge we have on genome evolution in Iridoviviridae, and in particular the function of conserved genes in this family. The manuscript reads well and while some parts are redundant and could benefit from a bit of trimming, it presents interesting avenues for future work and functional exploration of these genes. Additionally, even though most of the readers would have the basic cell biology background to understand the many acronyms used, I think some needs to be defined, especially when they represent important findings as opposed to commonly used chemicals or biological agents.

Response: We have gone through the manuscript and tried to reduce the number of acronyms.

Here are some minor editorial suggestions:

-Page 1, line 12: define ORF [while on line 29, add ORF after open reading frames]

Response: We have made these changes

-Page 1, line 17: define ER

Response: We made this change

-Page 3, line 110: I suggest “The cells were then incubated overnight at 4C with 2ml of block buffer…”

Response: Adjustment was made.

-Page 3, line 122: remove “then the cells”

Response: removed this phrase.

-Page 3, line 123: remove “and the cells were incubated”

Response: removed this phrase.

-Page 3, line 129-130: this sentence should be included in the Methods unless the format of a Brief Report is more flexible

Response: edited the first two sentences into one sentence.  Shortened the methods description with the sentence so that it now is an introduction to the gene that we will be exploring in this section.

-Page 4, line 141-143: these sentences are not necessary here and at best would need to be in the Methods, unless the format of a Brief Report is more flexible

Response: we have edited the sentence, so it is not a repetition of the materials and methods but serves as an introduction to the results for this particular gene.

-Page 4, line 145: To-PRO-3 should be defined in the Methods

Response: There is a brief section in the methods on TO-PRO-3 staining (lines 191-192).  We also noted in our reviewing this, that we had used different formats for spelling TO-PRO-3 in the manuscript.  We have gone through the entire document and fixed all references to TO-PRO-3 so it is consistent throughout the document

-Page 5, line 161: given the figure, I find “strong overlap” a bit exaggerated

Response: We have removed the word “strong” from our description.

-Page 5, line 178: replace the second “thiol oxidoreductases” by “which are”

Response: change made.

-Page 5, line 177-187: this whole section could be reduced as it goes into much details that are possibly out of scope for this Brief Report

Response: We have moved much of this material to the Introduction, where it is more appropriate. We have also tried to be non-repetitive with the Introduction.

-Page 5, line 199: “may not be limited”

Response: change made.

-Page 5, line 199-200: why is the basis for this particular function? Should be justified

Response: We have explained this as a theory and more work is needed to determine the actual function (lines 281-284).

-Page 6, line 201-204: these two sentences could be merged and shortened

Response: we have edited the sentences to be more concise and remove redundancy.

-Page 6, line 217: I suggest “in order to a successful infection of the host”

Response: adjustments were made to this sentence.

Round 2

Reviewer 1 Report

The revised manuscript, “Localization of Frog virus 3 conserved viral proteins 88R, 91R and 94L” by Penny and Brunetti describe experiments to determine the cellular localization of 3 ectopically expressed Frog virus 3 (FV3) proteins in baby green monkey kidney (BGMK) mammalian cells. This revision has been greatly improved and I have no further comments or suggestions.